# A Novel Bioactive Endodontic Sealer Containing Surface-Reaction-Type Prereacted Glass-Ionomer Filler Induces Osteoblast Differentiation

**DOI:** 10.3390/ma13204477

**Published:** 2020-10-09

**Authors:** Nobuyuki Kawashima, Kentaro Hashimoto, Masashi Kuramoto, Alamuddin Bakhit, Yasumiko Wakabayashi, Takashi Okiji

**Affiliations:** 1Department of Pulp Biology and Endodontics, Division of Oral Health Sciences, Graduate School of Medical and Dental Sciences, Tokyo Medical and Dental University (TMDU), 1-5-45 Yushima, Bunkyo-ku, Tokyo 113-8549, Japan; kawashima.n.endo@tmd.ac.jp (N.K.); k.hashimoto.endo@tmd.ac.jp (K.H.); m.kuramoto.endo@tmd.ac.jp (M.K.); yasumiko.en71.do346@gmail.com (Y.W.); 2Department of Endodontics, Faculty of Dentistry, King Abdulaziz University, Jeddah 21589, UAE; alam@bakhit.com

**Keywords:** S-PRG filler, endodontic sealer, osteoblasts, calcium-sensing receptor, MAP kinase

## Abstract

Surface-reaction-type prereacted glass-ionomer (S-PRG) fillers exhibit bioactive properties by the release of multiple ions. This study examined whether a novel endodontic sealer containing S-PRG fillers (PRG+) has the capacity to induce osteoblast differentiation. Kusa-A1 osteoblastic cells were cultured with extracts of PRG+, PRG− (an experimental sealer containing S-PRG-free silica fillers), AH Plus (an epoxy-resin-based sealer), and Canals N (a zinc-oxide noneugenol sealer). Cell viability and mineralized nodule formation were determined using WST-8 assay and Alizarin red staining, respectively. Osteoblastic-marker expression was analyzed with RT-qPCR and immunofluorescence. Phosphorylation of extracellular signal-regulated kinase (ERK) and p38 mitogen-activated protein kinase (MAPK) was determined with Western blotting. Extracts of freshly mixed PRG+, PRG−, and AH Plus significantly decreased cell growth, but extracts of the set samples were not significantly cytotoxic. Set PRG+ significantly upregulated mRNAs for alkaline phosphatase and bone sialoprotein (IBSP) compared to set PRG−, and upregulation was blocked by NPS2143, a calcium-sensing receptor antagonist. Set PRG+ significantly accelerated IBSP expression, mineralized nodule formation, and enhanced the phosphorylation of ERK and p38 compared with set PRG−. In conclusion, PRG+ induced the differentiation and mineralization of Kusa-A1 cells via the calcium-sensing receptor-induced activation of ERK and p38 MAPK.

## 1. Introduction

Root-canal filling is performed to attain a fluidproof seal throughout the root-canal system that was thoroughly cleaned and shaped with the intention of preventing the reinfection of the root-canal space. Endodontic sealers are used in conjunction with solid/semisolid core materials to obliterate irregularities between dentinal wall/tubules and core material, and are thus essential to enhance the three-dimensional sealing of complex root-canal systems [1,2]. Besides possessing this ability, an ideal sealer must offer several properties, including dimensional stability, proper handling, insolubility against tissue fluids, adhesion to canal walls, and biocompatibility. Endodontic sealers can be grouped on the basis of their primary constituents, such as zinc-oxide eugenol, calcium hydroxide, silicone, epoxy and methacrylate resins, and calcium silicates, although none of these is considered ideal [1,2].

One approach to improving endodontic sealers is to endow them with the ability to promote periapical tissue healing accompanied by mineralized tissue formation. In this regard, calcium-silicate-based sealers are promising because they exhibit good biocompatibility and osteogenic potential [3]. These properties are mainly attributed to their bioactivity, i.e., the ability to release calcium and hydroxyl ions from calcium hydroxide produced during the hydration of calcium silicates [4]; thus, the incorporation of bioactivity is a direction to explore in developing new classes of sealers that possess properties that enhance healing.

Surface-reaction-type prereacted glass-ionomer (S-PRG) fillers are a class of bioactive filler material produced by an acid-base reaction between porous silica glass-coated fluoroboroaluminosilicate glass fillers and a polyacrylic acid solution, by which a glass-ionomer phase is formed between the surface (silica glass) phase and the glass core phases [5]. S-PRG fillers have been incorporated in a number of dental materials, including resin composites [6,7] and a prototype root-canal sealer [8]. The bioactivity of S-PRG fillers can be attributed to their ability to release multiple ions, including boron, fluoride, silicate, and strontium [5], which may also provide the following biological reactions. S-PRG fillers inhibit demineralization and induce remineralization in the enamel and dentin because they release fluoride and its recharge [6,9]. S-PRG fillers promote the formation of mineralized apatite crystals via phosphoproteins [10], which support their ability to induce mineralization. Resin composites containing S-PRG fillers exhibit antibacterial activity mainly because of the release of fluoride ions and boric ions [11]. Moreover, recent studies demonstrated that an experimental pulp-capping material containing S-PRG fillers promotes the formation of this reparative dentin in exposed dental pulp in rats [7,12]. Compounds eluted from S-PRG fillers also promote osteoblastic differentiation in human mesenchymal stem cells [13]. On the basis of these findings, it is hypothesized that the incorporation of S-PRG fillers endows endodontic sealers with several beneficial properties, leading to proregenerative effects on periapical bone and cementum. However, little information is available regarding the effects of S-PRG fillers on osteoblast differentiation.

In this study, we examine whether a novel endodontic sealer containing S-PRG fillers could induce osteoblast differentiation. We also identify the signaling mechanisms responsible for the effects of the material on osteoblast differentiation.

## 2. Materials and Methods

### 2.1. Preparation of Sealer Extracts

An endodontic sealer containing S-PRG filler (prototype; Shofu, Kyoto, Japan; PRG+), an experimental endodontic sealer containing S-PRG-free silica filler (Shofu, Kyoto, Japan; PRG−), AH Plus (an epoxy resin-based sealer; Dentsply Sirona, Charlotte, NC, USA; AH), and Canals N (a zinc oxide-noneugenol sealer; Showa Yakuhin Kako, Tokyo, Japan; CN) were used in this study (Table 1). Each was mixed according to the manufacturers’ instructions, poured into a polypropylene container (8 mm in diameter, 3 mm high), and either placed immediately in culture medium (alpha MEM, 10 mL, FUJIFILM Wako Pure Chemical, Osaka, Japan; fresh group) or incubated at 37 °C under 100% humidity for five days to induce complete setting before immersion in the culture medium (set group). The culture medium containing the sealer was shaken at room temperature for 24 h and sterilized by passing it through a membrane filter (0.45 µm pore size, Sartorius, Göttingen, Germany).

Elemental analysis of each sealer extract was conducted with inductively coupled plasma atomic emission spectroscopy (ICP-AES: ICPS-8000, Shimadzu, Kyoto, Japan). The concentration of fluoride ions released from each set sealer sample was measured using a fluoride ion sensor (Horiba, Kyoto, Japan).

### 2.2. Cell Culture

Mouse osteoblastic cell line Kusa-A1 (RCB2081, Cell Engineering Division/RIKEN BioResource Research Center, Tsukuba, Ibaraki, Japan) was cultured in alpha MEM containing 10% fetal bovine serum (HyClone, Logan, UT, USA), and supplemented with antibacterial and antifungal agents (FUJIFILM Wako Pure Chemical, Osaka, Japan) at 37 °C, 5% CO_2_, and 100% humidity.

In the following experiments, cells were cultured in the presence or absence (control) of a 16-fold-diluted sealer extract. In some experiments, strontium ranelate (1 mM, LKT Laboratories, St Paul, MN, USA) or a calcium-sensing receptor (CaSR) antagonist (NPS2143-hydrochloride, 1 mM, Cayman, Ann Arbor, MI, USA) was added.

### 2.3. Cell-Viability Assay

Cells were seeded at 3 × 10^3^ cells/cm^2^ and cultured for 24, 48, or 72 h in the presence or absence (control) of a sealer extract. Cell viability was measured using a WST-8 assay (CCK-8, Dojindo Molecular Technologies, Kumamoto, Japan). Briefly, 10 μL of CCK-8 solution was added to each well of 96-well plate (100 μL of culture medium per well), which was further incubated for one hour. Then, absorbency was measured at 450 nm optical density using a microplate reader (Sunrise, Tecan, Männedorf, Switzerland).

### 2.4. Nodule Formation

Cells were seeded at 5 × 10^4^ cells/cm^2^ and cultured with or without a sealer extract in an osteogenic medium containing 5 mM beta-glycerol phosphate (FUJIFILM Wako Pure Chemical, Osaka, Japan) and 0.2 mM L-ascorbic acid-2-phosphate (FUJIFILM Wako Pure Chemical, Osaka, Japan). Mineralized nodules were fixed with methanol and stained with Alizarin red S (FUJIFILM Wako Pure Chemical, Osaka, Japan). The density of mineralized nodules was measured with ImageJ software (ver 1.53e, U. S. National Institutes of Health, Bethesda, MD, USA).

### 2.5. Reverse Transcription-Quantitative Polymerase Chain Reaction

Cells were seeded at 1 × 10^4^ cells/cm^2^ and cultured with or without a sealer extract for 48 h. RNA was extracted using a QuickGene isolation kit (Kurabo, Osaka, Japan), and cDNA was generated from the extracted RNA using reverse transcriptase (RevertAid H minus reverse transcriptase, Thermo, Waltham, MA, USA). Reverse transcription-quantitative polymerase chain reaction (RT–qPCR) was performed on the cDNA with Taq polymerase containing SYBR green (GoTaq qPCR Master Mix, Promega, Madison, WI,USA) and specific primers for alkaline phosphatase (ALP: upper, 5′-GATTACGCTCACAACAACTACCAG-3′, lower, 5′-GGAATGTAGTTCTGCTCATGGAC-3′, NM_007431, 140 bps, Eurofins Genomics, Tokyo, Japan) and bone sialoprotein (Ibsp: upper, 5′-TATGAAGTCTATGACAACGAGAACG-3′, lower, 5′-AGTAATAATTCTGACCCTCGTAGCC-3′, NM_008318, 121 bps, Eurofins Genomics, Tokyo, Japan), using an RT-PCR detection system (CFX96, Bio-Rad, Hercules, CA, USA). Beta-actin (BA: upper, 5′-GTAAAGACCTCTATGCCAACACAGT-3′, lower, 5′-AATGATCTTGATCTTCATGGTGCTA-3′, NM_007393, 122 bps, Eurofins Genomics, Tokyo, Japan) was used as internal control.

### 2.6. Immunofluorescence

Cells were seeded at 1 × 10^4^ cells/cm^2^ and cultured with or without a sealer extract for 48 h in cell-culture chamber slides (Fukae Kasei Co. LTD, Kobe, Japan). Immunofluorescence staining was performed on fixed cells with 4% paraformaldehyde in phosphate-buffered saline for 12 h at 4 °C. Briefly, following normal goat serum incubation to prevent nonspecific blocking, a rabbit anti-bone sialoprotein (IBSP) antibody (1:1000, #5468, Cell Signaling Technology, Danvers, MA, USA) was applied to the samples and incubated for 24 h at 4 °C. Goat antirabbit-IgG conjugated with Alexsa488 (1:500, Molecular Probes/Thermo fisher, Waltham, MA, USA) was used as a secondary antibody, and 4′,6-diamidino-2-phenylindole (DAPI, Dojindo Molecular Technologies, Inc, Kumamoto, Japan) was used as nuclear staining. The Axio Observer fluoresce microscope (Zeiss Microscopy, Jena, Germany) was used for observation and taking photos.

### 2.7. Western Blotting

Cells were seeded at 1 × 10^4^ cells/cm^2^, cultured for 24 h, and lysed with a radioimmunoprecipitation assay buffer containing proteinase and phosphatase inhibitors (cOmplete and PhosSTOP; Sigma–Aldrich, St. Louis, MO, USA). Lysates were applied to a 10% polyacrylamide gel, and the proteins were separated using sodium dodecyl sulfate-polyacrylamide gel electrophoresis. Then, proteins in the polyacrylamide gel were transferred to polyvinylidene fluoride membranes (GE Healthcare, Chicago, IL, USA). Membranes were incubated with 1:1000-diluted polyclonal antibodies against extracellular signal-regulated kinase (ERK), phosphorylated ERK (pERK; 1:1000; BD Bioscience, San Jose, CA USA), p38 mitogen-activated protein kinase (MAPK), and phosphorylated p38 MAPK (pp38; BD Bioscience) overnight at 4 °C. Then, membranes were incubated with horseradish peroxidase-labeled antirabbit IgG (1:5000, BioRad, Hercules, CA, USA) at room temperature for one hour. A horseradish peroxidase substrate (Immobilon Forte Western HRP substrate, Merck, Kenilworth, NJ, USA) was applied as the chemiluminescent reagent, and the developed membranes were analyzed using an LAS-3000 Mini Image Analyzer (Fuji Film, Tokyo, Japan).

### 2.8. Statistical Analysis

Data were evaluated by one-way analysis of variance (ANOVA) followed by the Tukey–Kramer test (for cell growth) or the Student’s *t*-test (for osteoblastic-marker expression and mineralized-nodule formation) using Prism 6 software (GraphPad, San Diego, CA, USA). The *p* values less than 0.05 were regarded as statistically significant.

## 3. Results

### 3.1. Element/Ion Concentration of Sealer Extracts

First, the concentration of elements/ions released from fresh and fixed samples was measured using ICP-AES and a fluoride ion sensor. As shown in Table 1, concentrations of B, Al, Si, and F- were higher, and that of Sr was lower in the fresh PRG+ extract compared with the set PRG+ extract. Moreover, B and Al were detected only in PRG+. CN released Si, Na, and Zn, and AH released Si and Na.

### 3.2. Cell Growth

Next, the effects of sample extracts on cell growth were determined. Fresh PRG+ and AH extracts resulted in significantly lower cell growth at 48 h compared with the control, and cell growth with the fresh AH extract was significantly lower than that with PRG+ or PRG− over 72 h (*p* < 0.05; Figure 1A). None of the set extracts from the sealers was associated with significant cytotoxic effects (*p* > 0.05; Figure 1A).

### 3.3. Osteoblastic-Marker Expression and Mineralized-Module Formation

Then, the effects of PRG+ and PRG− extracts on osteoblastic-marker expression and mineralized-nodule formation were evaluated. Kusa-A1 cells cultured in the presence of the set PRG+ extract exhibited elevated expression of ALP and IBSP mRNA (*p* < 0.05; Figure 1B), and the IBSP protein (Figure 1C), and increased nodule formation (*p* < 0.05; Figure 1D) compared with those cultured with the set PRG− extract. No mineralized nodule was formed in the cells without culture in the osteogenic medium.

### 3.4. Effect of CaSR Antagonist on Osteoblastic-Marker Expression, and ERK and p38 Phosphorylation

The ICP-AES experiment revealed that one of major elements released from set PRG+ was strontium, which activates intracellular signaling via CaSR [14,15]; thus, the effect of a CaSR antagonist on osteoblastic-marker expression in PRG+ extract-treated Kusa-A1 cells was examined. The CaSR antagonist significantly downregulated ALP and IBSP mRNA expression in Kusa-A1 cells cultured with the set PRG+ extract (*p* < 0.05; Figure 2A).

Next, the activity of ERK and p38 signaling, known as the downstream targets of CaSR [14,15], was investigated in Kusa-A1 cells treated with PRG+, PRG− and Sr^2+^. Kusa-A1 cells exposed to the set PRG+ extract exhibited higher phosphorylation of ERK and p38 compared with that of the set PRG− extract (Figure 2B). The CaSR antagonist downregulated the phosphorylation of ERK in Kusa-A1 cells exposed to the set PRG+ extract (Figure 2B). Sr^2+^ also induced the phosphorylation of ERK and p38, which was downregulated by the CaSR antagonist (Figure 2B). α-Tubulin was used as loading control.

## 4. Discussion

In this study, root-canal sealers were evaluated for their cytotoxicity and ability to induce osteoblastic differentiation in Kusa-A1 cells, which were established from mouse bone-marrow stromal cells [16] and exhibited properties of mature osteoblasts [17]. Kusa-A1 cells were used for the evaluation of osteoblastic toxicity [18].

The cell-growth test demonstrated that, at the used extract concentrations, extracts from the set samples did not have any inhibitory effects on Kusa-A1 cells (Figure 1A); thus, this concentration was used for analysis of osteoblastic differentiation. However, extracts from fresh PRG+, PRG−, and AH exhibited cytotoxicity (Figure 1A). The cytotoxic effect of fresh AH was consistent with previous reports and most likely caused by the release of free epoxy resin monomers before completion of the setting reaction [19]. The good biocompatibility of CN was also consistent with a previous report [20]. Cytotoxic effects associated with fresh PRG+ and PRG− extracts were significantly weaker than that of the fresh AH extract (Figure 1A). PRG+ is a hybrid cement material that sets by two acid-base reactions of polycarboxylic acid sodium salt with either S-PRG fillers or zinc oxide-based inorganic compound fillers. PRG− sets with only the latter reaction; thus, the cytotoxicity of PRG+ and PRG− may be related to the cytotoxic effect of polycarboxylate and glass-ionomer cements [21]. Elemental analysis revealed that boron and aluminum ions were released from fresh PRG+, but not from fresh PRG− (Table 1). These ions are known to be cytotoxic [22,23], but it is unlikely that they are primarily responsible for the cytotoxicity associated with PRG+ because fresh PRG− exhibited a similar level of cytotoxicity.

The release of strontium from set PRG+ but not from set PRG− (Table 1) is of interest, as strontium promotes the osteogenic differentiation/mineralization of dental papillae cells [14]. Strontium detected in the set PRG+ sample may have originated from the glass-ionomer phase formed during the setting reaction, in which phase a considerable amount of Sr released from acid-eroded glass core of S-PRG fillers may have become trapped. In addition, both PRG+ and PRG− contain strontium compounds (indicated as “additive” in Table 1) as setting regulators, which turn into insoluble salt after setting and thus contribute to the strontium elution of only fresh samples.

Osteoblast differentiation was evaluated by the expression of ALP and IBSP, which are important markers for osteoblast activity during early osteoblast differentiation [24]. ALP initiates the calcification process by providing inorganic phosphates [25], and IBSP is associated with mineral nucleation in the matrix of primary membrane bone [26]. Results also showed that the set PRG+ extracts promoted differentiation and mineralization in Kusa-A1 cells (Figure 1B–D). Among multiple ions released from set PRG+ (Table 1), strontium ions may have had a strong effect because strontium ranelate was shown to promote odontoblast differentiation and mineralization in vitro, and the direct application of strontium ranelate to pulp tissue was found to induce the formation of osteodentin-like mineralized tissue [14]. Furthermore, strontium ranelate also induces osteoblastic differentiation in primary murine bone cells [27] and mesenchymal stem cells [28]. Strontium binds to CaSR, and activated CaSR induces the upregulation of several intracellular signaling pathways, including Akt [14] and Erk1/2 [15]. The phosphorylation of ERK and p38 MAPK signaling was induced by strontium via CaSR activation (Figure 2B), which may be a mechanism of the action of strontium and could be associated with periapical-tissue healing, as is discussed below. Moreover, as boron ions were reported to induce osteoblastic differentiation in stem cells from exfoliated deciduous teeth [23], they may also contribute to the observed PRG+-induced differentiation and mineralization of Kusa-A1 cells. Further study is necessary to reveal the whole function of ions/compounds released from PRG+ sealer in osteoblast differentiation/mineralization.

There are numerous membrane-bound ion channels involved in important physiological processes. Among these channels is the CaSR, which is a class C G-protein-coupled receptor [22] that is expressed in various cells including osteoblasts [29]. CaSR signaling in osteoblasts is essential for their differentiation and mineralization ([30]]. Results of this study revealed that NPS-2143, a potent CaSR antagonist, down-regulated the osteoblastic gene expression induced by the PRG+ extract in Kusa-A1 cells (Figure 2A). This finding suggests that ions released from PRG+ induce osteoblastic differentiation via the CaSR. Previous studies have shown that strontium ions activate CaSR-dependent signaling in HEK293 cells in a manner similar to calcium ions [31]. Furthermore, CaSR is known to be involved in strontium ranelate-induced osteoblast differentiation and mineralization [32].

CaSR was also linked to several signaling pathways such as EK1/2 [33,34] that are essential for osteoblast differentiation [35]. Moreover, CaSR activation induces the phosphorylation of p38 [36], and the activation of p38 is required for osteoblast differentiation [37]. The results of this study revealed that phosphorylation of ERK and p38 induced by set PRG+ was blocked by the CaSR antagonist (Figure 2B). Strontium-ranelate-induced phosphorylation of ERK, and p38 was also downregulated by the CaSR antagonist. Moreover, the CaSR antagonist blocked the mRNA expression of ALP and IBSP genes, and was upregulated by set PRG+. Taken together, our findings indicate that CaSR activation via ions released from set PRG+, especially strontium ions, and the subsequent activation of ERK and p38 signaling are involved in the observed osteoblastic differentiation and mineralization of Kusa-A1 induced by set PRG+. However, the involvement of signaling pathways independent of ERK and p38, such as the calcineurin/nuclear factor of activated T-cell or Wnt/beta-catenin signaling, or the activation of other ion channels cannot be ruled out.

## 5. Conclusions

In conclusion, under the present experimental conditions, PRG+ only showed cytotoxicity in the fresh state, and induced the differentiation and mineralization of Kusa-A1 involving the activation of the ERK and p38 signaling pathways via CaSR. These findings suggest that PRG+ is a promising new class of bioactive root-canal sealer that has the ability to promote periapical-tissue healing and mineralized-tissue formation. However, it is necessary to evaluate several other properties of the material, including sealing ability, structural stability, and other physical properties, before its clinical application.

## Figures and Tables

**Figure 1 materials-13-04477-f001:**
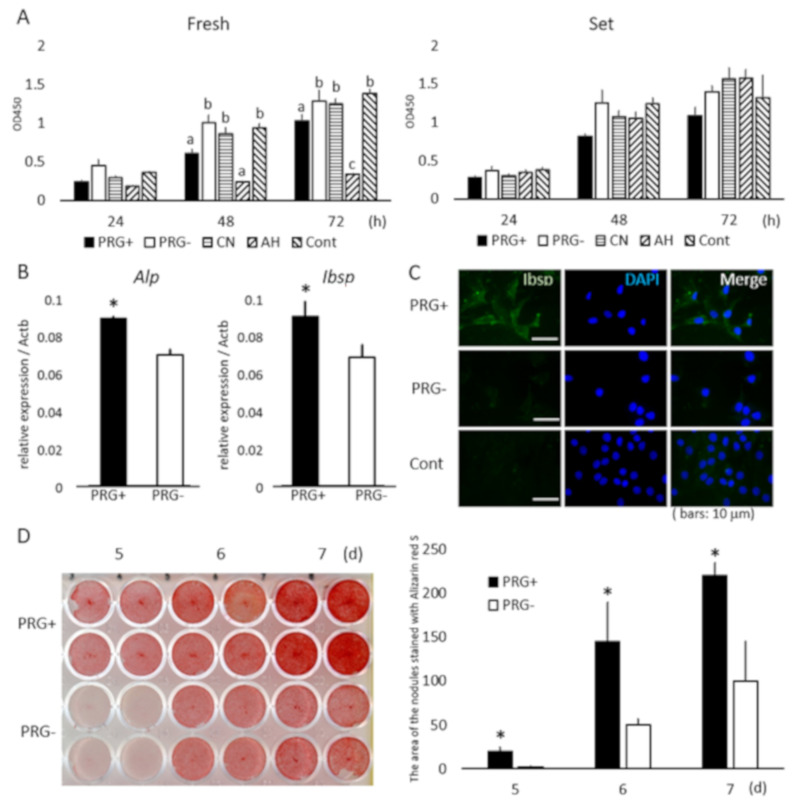
(**A**) Cytotoxicity of fresh and set extracts to Kusa-A1 cells. (**A**) Fresh PRG+ and AH resulted in significantly lower cell growth at 48 and 72 h compared with PRG−, CN, and control (*p* < 0.05). Absorbance in each well following incubation with CCK-8 measured at 450 nm optical density (OD). AH resulted in significantly lower cell growth at 72 h than PRG+ and PRG− did (*p* < 0.05). Different letters at the same time point represent statistically significant differences (*p* < 0.05); *n* = 4. Cell growth was similar in the presence of all set samples; differences were not statistically significant (*p* < 0.05); *n* = 4. (**B**) mRNA expression of alkaline phosphatase (ALP) and bone sialoprotein (IBSP) in Kusa-A1 cells promoted by set PRG+ extract (*p* < 0.05); *n* = 8; **p* < 0.05. (**C**) Immunofluorescence staining revealed enhanced expression of IBSP in Kusa-A1 cells cultured with PRG+ extract. Rabbit IgG was used as first antibody instead of anti-IBSP antibody in control samples (Cont); *n* = 3. Bars indicate 10 μm. DAPI (4′,6-diamidino-2-phenylindole) was used for nuclear staining. Combined pictures with IBSP and DAPI indicated as Merge. (**D**) Mineralized-nodule formation demonstrated with Alizarin red staining followed by densitometric analysis. Mineralized-nodule formation upregulated in Kusa-A1 cells in the presence of set PRG+ extract; *n* = 4; **p* < 0.005.

**Figure 2 materials-13-04477-f002:**
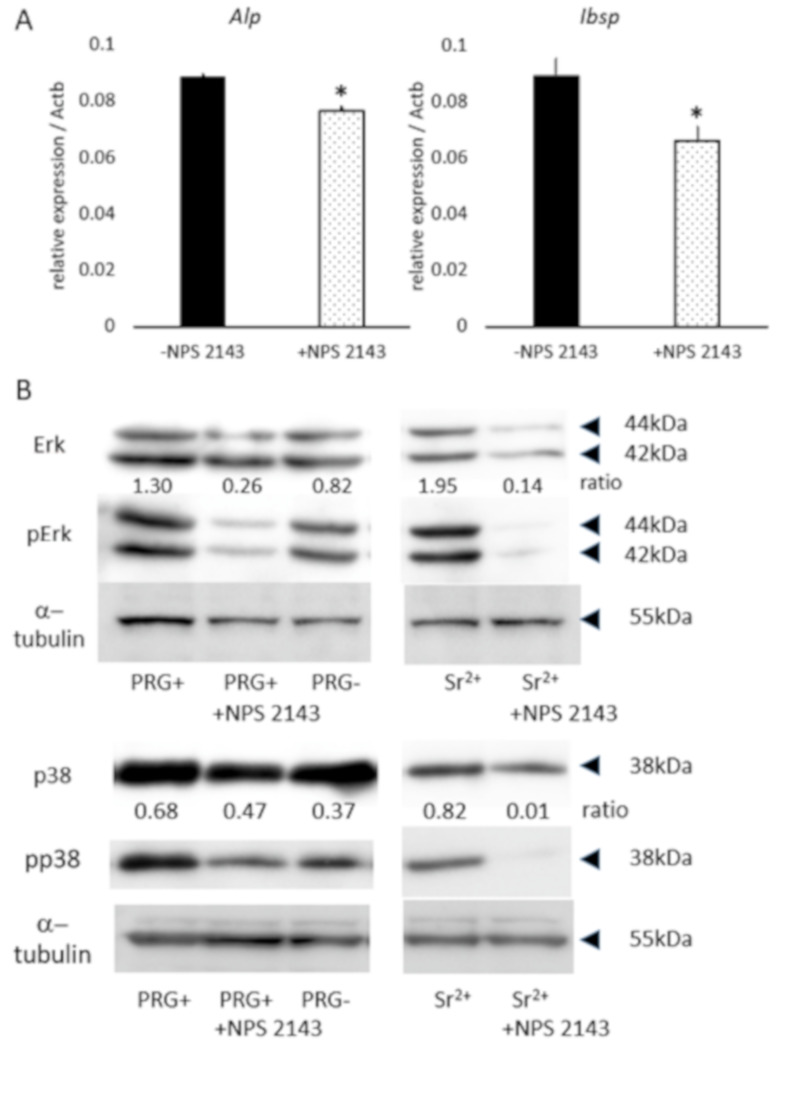
(**A**) mRNA expression of ALP and IBSP induced by set PRG+ extract significantly downregulated by application of NPS2143 (CaSR antagonist) in Kusa-A1 cells; **p* < 0.05. (**B**) Phosphorylation of extracellular signal-regulated kinase (ERK) and p38 promoted by set PRG+ extract is downregulated by the application of NPS2143. Phosphorylation of ERK and p38 promoted by Sr^2+^ is also downregulated by NPS2143 application. α-Tubulin was used as loading control. Experiments were repeated three times with similar results.

**Table 1 materials-13-04477-t001:** Material properties and concentrations of elements and ions released from fresh and set extracts.

Materials	Main Composition	Elements/Ion Concentration (ppm)							
			B	Al	Si	Na	Zn	Sr	F^-^
Prototype endodontic sealer containing S-PRG filler (Shofu; PRG+)	Powder: zinc oxide-based inorganic compound filler, S-PRG filler *, additive Liquid: poly carboxylic acid sodium salt, water, other	Fresh	236	28	42	240	61	27	236
		Set	198	0	2	240	46	47	76
Experimental endodontic sealer containing S-PRG-free silica filler (Shofu; PRG-)	Powder: zinc oxide-based inorganic compound filler, silica-filler, additive Liquid: poly carboxylic acid sodium salt, water, other	Fresh	0	0	53	480	1024	28	199
		Set	0	0	4	400	57	0	154
Canals N (Showa Yakuhin Kako; CN)	Powder: zinc oxide, barium sulfate, bismuth carbonate oxide, rosin Liquid: fatty acids, propylene glycol	Fresh	0	0	7	120	29	0	0
		Set	0	0	10	100	16	0	0
AH Plus (Dentsply Sirona; AH)	Paste A: bisphenol-A epoxy resin, bisphenol-F epoxy resin, calcium tungstate, zirconium oxide, iron oxide pigments, Aerosil Paste B: N-dibenziyl-5-oxanonane, aminoadamantane, tricyclodecane-diamine, calcium tungstate, zirconium oxide, Aerosil, silicone oil	Fresh	0	0	28	320	0	0	0
		Set	0	0	42	660	0	0	0

* surface reaction type pre-reacted glass-ionomer filler-based fluoroboroaluminosilicate glass.

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
