# Peer review of "A Novel Bioactive Endodontic Sealer Containing Surface-Reaction-Type Prereacted Glass-Ionomer Filler Induces Osteoblast Differentiation"

_materials, 2020, doi:10.3390/ma13204477_

Round 1
Reviewer 1 Report
In the manuscript by Kawashima et al. the authors investigated the effect a novel bioactive endodontic sealer containing a surface-reaction-type pre-reacted glass-ionomer filler (S-PRG) (PRG+) on osteoblast differentiation and compared it to an experimental sealer containing S-PRG-free silica fillers (PRG-). First they showed that while medium incubated with fresh sealers (PRG+ and PRG-, respectively) had a slightly toxic effect on the growth of Kusa-A1 cells, set sealers incubated medium had no influence on cell proliferation. However, medium incubated with set PRG+ increased osteoblast differentiation, Alp and Ibsp expression and Erk and p38 phsophorylation. The authors suggest that the latter might be caused by strontium released by PRG+ into the medium, as the addition of strontium ranelate to the medium had a similar result. All these effects might by mediated by the membrane-bound ion channel CaSR, as a CaSR antagonist led to the down-regulation of the PRG+ mediated increase in Alp and Ibsp expression and Erk and p38 phosphorylation.
This is an interesting work evaluating the potential of a new bioactive root canal sealer. Although it is shown that PRG+ in its set state is not cytotoxic and that it can induce differentiation and mineral deposition of osteoblasts, I have my concerns regarding too preliminary and lacking controls. Therefore I recommend publication after major revisions. For details see below.
- I am wondering why in the fresh state, both PRG+ and PRG- release similar amounts of Sr into the medium, but in their set state PRG+ even releases more (nearly double the amount), while PRG- none? That does not make sense to me.
- For me one of the greatest drawbacks of this study is, that while in Fig. 1A control medium and media incubated with CN and AH were used, in all other figures only PRG+ and PRG- conditioned media are used. I think it is also necessary to use control medium for the other experiments, as PRG- medium might have negative effects on osteoblast differentiation, gene expression,..., as it contains more Na and F than the PRG+ medium.
-) The result section is very minimalistic and should be re-written in my opinion. I am missing a flow from one chapter to the other with a rational explanation for the reasons why the individual experiments were performed. For example, there is no reasonable explanation, why especially a CaSR antagonist was used. Have also other compounds affecting osteoblast differentiation been tested? Especially in the experiments using NPS2143 controls with normal medium are important.
- The title is too strong for me. Blocking the respective pathways may in any case counteract the effects seen by PRG+, independently if PRG+ acts through these pathways or not.
- Figure 1A: Not all symbols in the legend are present on the graph and vice versa. Also the presentation of the statistical significance with the small letters is confusing.
- Figure 1F: The label for the y axis should be done as in C and D. Also the p value for statistical significance should be in the figure legend and not in the figure.
- Figure 2A: Is there no standard deviation? How many times has the experiment been repeated? Also the p value for statistical significance should be in the figure legend and not in the figure.
- Figure 2B: A loading control (e.g. actin) is missing. SrRn instead of StRn in some cases (also in the figure legend). Since the experiments were repeated three times with similar results statistics over the ratios between total and phosphorylated Erk and p38, respectively, should be included. I am wondering that the ratio (Erk) in the case of PRG+ is 1.15 and in the case of SrRn only 0.96.
- In M&M superscript should be used for cell numbers (e.g. 3 x 103 instead of 3 x 103).
Reviewer 2 Report
The Authors with this study would like to demonstrate that a novel endodontic sealer containing S-PRG fillers (PRG+) may exert the capacity to induce osteoblast differentiation. With their analyses they like to demonstrate that using a surface-reaction-type pre-reacted glass-ionomer (S-PRG) induced differentiation and mineralization of Kusa-A1 cells via calcium-sensing receptor-induced activation of Erk and p38 MAPK.
The aim is of interest but the demonstration lacks of several other analyses, mainly regarding the osteoblastic differentiation and function.
Main pitfalls:
1) The Title is extremely long and dispersive. It must be consistently shortened and focused on what the Authors demonstrate using the minimum of words;
2) The analyses aimed at demonstrating the osteoblastic differentiation need do be consistently improved: first of all no cells are observable. How is possible to write that an osteoblastic differentiation has happened without showing cells that become osteoblasts at least with IF analyses, positive for some of the osteogenic markers?
Secondly: osteogenic markers are NOT only BSP and ALP (the second one is often aspecific) but many others. The authors must show either by RT-PCR and by WB and/or IF how during the time (7,14 and 21 days) the positivity for all the osteogenic markers (including ospeopontin, ostonectin, osteocalcin, RUNX-2 and Ostrix (transcription factors)) changes.
All the previous is the minimum requirement in order to establish that the osteogenesis and osteoblastic differentiation is happening.
The remaining part of the study is well done with antagonists.
Reviewer 3 Report
Ref: materials-922098
Title: A Novel Bioactive Endodontic Sealer Containing 2 Surface-reaction-type Pre-reacted Glass-ionomer 3 Filler Induces Osteoblast Differentiation via 4 Extracellular Signal-regulated Kinase and p38 5 Mitogen-activated Protein Kinase Pathways.
This is a very interesting study regarding the effect of a S-PRG filler-containing endodontic sealer on differentiation of osteoblasts in teeth. Here are some points that should be addressed to improve the paper.
- Please reduce the title of the manuscript.
- Please give more information for the mechanism of action of strontium and boron ions and how they promote periapical tissue healing along with mineralized tissue formation. Please provide more references.
- Is there any information for the composition of S-PRG fillers? Are these concentrations of the strontium and boron anions enough to achieve therapeutic effects in clinical conditions?
Reviewer 4 Report
The authors tested whether a novel endodontic sealer containing S-PRG fillers (PRG+) has the capacity to induce osteoblast differentiation using Kusa-A1 cells compared with PRG- (an experimental sealer containing S-PRG-free silica fillers), AH Plus, and Canals N. Their experiments showed that extracts of freshly-mixed PRG+, PRG- and AH Plus significantly decreased cell growth, but extracts of set samples were not significantly cytotoxic. Set PRG+ significantly upregulated mRNAs for alkaline phosphatase and bone sialoprotein compared with set PRG-, and the upregulation was blocked by NPS2143. Set PRG+ significantly accelerated mineralized nodule formation and enhanced phosphorylation of Erk and p38 compared with set PRG-. They concluded that PRG+ induced differentiation and mineralization of Kusa-A1 cells via calcium-sensing receptor-induced activation of Erk and p38 MAPK. The experiments seemed well performed. The manuscript was also well written. Here are my a few concerns and suggestions:
1: mRNAs of alkaline phosphatase and bone sialoprotein were examined by PCR. However, the authors did not discuss what were the roles of alkaline phosphatase and bone sialoprotein in osteoblast differentiation/mineralization?
2: Extracts of PRG+ and PRG- contained various elements as showed in Table 1. The authors observed that PRG+ and PRG- had different roles in differentiation of osteoblasts and speculated that strontium and boron were the key elements for the differentiation of osteoblasts. However, there was no direct evidence although some references were cited to support their hypotheses. It is highly recommended that the authors test these hypotheses in their system.
Round 2
Reviewer 1 Report
Although the authors replied to most of my comments more or less satisfactorily and the paper was improved, I still have some minor comments/concers:
-) Regarding the release of Sr from into the medium: I still don't understand the differences between PRG+ and PRG- regarding the "fresh" and "set" state. In the reply it was stated "Following setting of PRG-, strontium salts in the additive had been reacted with polycarboxylic acid to form insoluble salts. Therefore, no release of strontium was observed in set PRG- samples. All the strontium ions detected in the extracts of set PRG+ were released from PRG fillers." However, polycarboxylic acid was also present in PRG+. And this also does not explain to me, why there is more Sr in the "set" than in the "fresh" PRG+ sealer, although they were both incubated for the same time period in medium. Please explain better in the paper, as also other readres might wonder.
-) Figure 1A: This presentation is much clearer. However, the symbols for PRG- and CN are the same (although the bars are different). And I just realized that the labelling of the y-axes "OD450" is nowhere explained why it refelcts cell growth.
- Figure 1B: Which graph presents Alp and which Ibsp (labelling missing)?
- Figure 1: Although a control is now shown at least for the immunofluorescence of Ibsp, I would have liked to see also a control for the differentiation experiment. As there is already at least Alp and Ibsp expression in the uninduced cells incubated with PRG+, is there also Alizarin red staining detectable in uninduced cells? Therefore, it would also be good to show AR staining for uninduced cells (day 0). And I am also wondering why there is nearly no Ibsp staining detectable in the PRG-sample, although the expression is only increased by 15-20% by PRG+?
- Chapter 3.2 first sentence: "Next" instead of "Nest"
- "Next, the activity of Erk and p38 signaling, known as dowstream of CaSR [14, 15], was investigated in kusa-A1 cells treated with PRG+, PRG- and strontium ranelate." I think "targets" is missing after "downstream" and "Kusa-A1" instead of "kusa-A1".
- Figure 1B: No explanation was given why the ratio between pErk to total Erk in the case of Sr+ is only 0.96, although the density of the pERK bands is much stronger than that of the total Erk? Since these experiments were repeated three times, this should be shown in a graph with standard deviations and statistical significance and not just stated "with similar results".
Reviewer 2 Report
The osteogenic demonstration is weak. In addition no IF analyses were performed in order to demonstrate that cells undergo osteogenic differentiation.
Reviewer 4 Report
No further comments.
Author Response
Thank you for your kind suggestions.